# Design and Experimental Assessment of Real-Time Anomaly Detection Techniques for Automotive Cybersecurity

**DOI:** 10.3390/s23229231

**Published:** 2023-11-16

**Authors:** Pierpaolo Dini, Sergio Saponara

**Affiliations:** Department of Information Engineering, University of Pisa, Via Girolamo Caruso n.16, 56100 Pisa, Italy

**Keywords:** artificial intelligence, machine learning, statistical learning, controlled area network, networking, cybersecurity, automotive, mechatronics

## Abstract

In recent decades, an exponential surge in technological advancements has significantly transformed various aspects of daily life. The proliferation of indispensable objects such as smartphones and computers underscores the pervasive influence of technology. This trend extends to the domains of the healthcare, automotive, and industrial sectors, with the emergence of remote-operating capabilities and self-learning models. Notably, the automotive industry has integrated numerous remote access points like Wi-Fi, USB, Bluetooth, 4G/5G, and OBD-II interfaces into vehicles, amplifying the exposure of the Controller Area Network (CAN) bus to external threats. With a recognition of the susceptibility of the CAN bus to external attacks, there is an urgent need to develop robust security systems that are capable of detecting potential intrusions and malfunctions. This study aims to leverage fingerprinting techniques and neural networks on cost-effective embedded systems to construct an anomaly detection system for identifying abnormal behavior in the CAN bus. The research is structured into three parts, encompassing the application of fingerprinting techniques for data acquisition and neural network training, the design of an anomaly detection algorithm based on neural network results, and the simulation of typical CAN attack scenarios. Additionally, a thermal test was conducted to evaluate the algorithm’s resilience under varying temperatures.

## 1. Introduction

### 1.1. Motivations for CAN Cybersecurity

Rapid technological progress has made it possible for a wide range of industries, such as robotics, mechatronics, automation, and the automotive sector, to operate in a highly networked global environment. Although efficiency has increased significantly, these domains are now more vulnerable to growing cybersecurity threats [1,2,3,4,5]. The extensive usage of automation systems in the field of robotics has made them susceptible to cyber-attacks, which might endanger both human safety and valuable assets [6,7,8,9]. Comparably, industries across the board are now vulnerable to cybersecurity attacks due to the incorporation of software components in mechatronics, which combines electronics, software, and mechanics [10,11,12,13]. Automation in digital networks across several industries has increased the risk of cyber-attacks, which might have major operational and financial consequences [14,15,16,17]. The continuous shift in the automotive industry toward connected and automated vehicles has highlighted how important cybersecurity is for protecting user privacy and vehicle management systems. In digital technology-driven areas, cybersecurity essentially acts as the cornerstone for safeguarding against monetary losses and guaranteeing human welfare. This requirement also applies to car networking systems, where hackers may put human lives in danger. The automotive sector actively works with cybersecurity professionals to strengthen security measures. They concentrate on secure updates, customized communication protocols, and intrusion detection systems for vehicular networks. To sum up, cybersecurity plays a critical role in these many industries, acting as the foundation for guaranteeing safety and smooth functioning. Through ongoing research and innovation, we can create a future where technology is strong and resilient against cyber-attacks, ensuring security and peace of mind.

In modern vehicles, there are numerous electronic control units (ECUs) used for automation and comfort features, both of the driver and passengers [18]. Included in this class are ECUs on which advanced algorithms and features are integrated for cruise control, airbag control, temperature management, assisted parking, infotainment, etc. One of the problems for security in data exchange and enabling cybersecurity issues is related to the interconnections between the various ECUs [19]. The use of remote access points, including Wi-Fi, USB, Bluetooth, 4G/5G, and OBD-II interfaces, has increased dramatically in the automotive industry in recent years [20]. On the other hand, because of their widespread use, automotive networks are now more susceptible to outside attacks. These networks may be targeted by hostile parties who want to take over, change car systems, or steal confidential data. As a result, creating strong security systems that can identify and stop such breaches or assaults is imperative. In the automotive industry, one of the most often used protocols for intra-ECU communication is the Controller Area Network (CAN) bus. The numerous noteworthy characteristics of the CAN protocol include its ease of wiring, strict response times, high immunity to interference, error containment, and multi-master protocol capabilities [21]. The Carrier Sense Multiple Access/Bit-wise Arbitration (CSMA/BA) technique is used by the CAN system to control bus traffic. When two or more nodes initiate transmission simultaneously, an arbitration process based on ID prioritizing is commenced. But it is important to understand that the CAN bus protocol does not include a basic security mechanism, which leaves connected devices vulnerable to adversaries [22]. By taking advantage of weaknesses in the CAN bus protocol, aggressors can initiate various assaults that have the potential to impede vehicle functions. The lack of an authentication element in CAN frames creates this vulnerability, which allows any transmitting ECUs to mimic other ECUs. Additionally, the lack of content encryption in CAN frames gives adversaries a simple way to examine how target ECUs operate using CAN frame records from the past [23].

Although data encryption techniques have been proposed in the literature, their application to the CAN bus protocol has shown them to be unsuccessful [24]. Moreover, frames with lower IDs (the highest priorities) might preempt the bus using the priority-based arbitration process, forcing all other CAN frames to cede. In light of these technical aspects, a wide range of techniques have been put forth in the body of literature to detect possible assaults on the CAN network. Our novel method, which we provide in this study, improves and applies previous research findings to real-world CAN networks.

Our approach makes use of real-time analysis on a simple platform, and thermal testing is used to validate the results. The first part will address data acquisition, explaining the methods and approaches used to collect and organize data. Because of the remarkable multi-class classifier performance of Artificial Neural Networks (ANNs) and their simplicity in embedded system integration, we have decided to use them. The results of experimental testing, which include the evaluation of several attack scenarios put out in the literature to verify the Intrusion Detection System (IDS) algorithm, will then be presented. In the end, the robustness of our proposed method with respect to temperature variation is shown, and this is well known to affect the circuit aspects, and consequently, the physical layer associated with the CAN protocol.

### 1.2. The State-of-the-Art on CAN Cybersecurity

There are several detection algorithms proposed in the literature to address the cybersecurity issues related to the Controller Area Network (CAN) protocol. Here is an overview of some of the state-of-the-art detection algorithms.
(1)Two-Step Algorithm: This algorithm uses a mixed approach of temporal-spatial analysis to detect cyber-attacks over the CAN bus. The algorithm first detects the abnormal behavior of the CAN bus and then identifies the source of the attack [25,26,27,28,29].(2)Intrusion Detection System (IDS): IDS is a popular security solution that uses crypto- graphic-based software to address CAN network security issues. The IDS ensures that the exchanged CAN data frame between the two end nodes is authorized. Researchers have proposed various IDS algorithms, such as a lightweight algorithm based on the observance of CAN packets frequencies, an anomaly-based detection method based on the time interval feature of the consecutive CAN packets, and a graph-based feature method that uses machine learning algorithms [30,31,32,33,34,35,36,37,38,39,40,41,42].(3)CAN-ADF: The Controller Area Network Attack Detection Framework (CAN-ADF) is a framework that uses field classification, modeling, and anomaly detection to detect cyber-attacks on unknown CAN bus networks. The framework uses a holistic approach to detect cyber-attacks and provides a comprehensive solution to the cybersecurity issues related to the CAN protocol [43,44].(4)Deep Learning Techniques: Intrusion Detection Systems (IDSs) using deep learning techniques are also proposed in the literature. These IDSs identify cyber-attacks when given a sample of network traffic collected from real-world computer networks. The IDSs using deep learning techniques are powerful and can detect cyber-attacks with high accuracy [45,46,47,48,49].
In summary, various detection algorithms have been proposed in the literature to address the cybersecurity issues related to the CAN protocol. These algorithms use different approaches, such as temporal-spatial analysis, cryptographic-based software, anomaly-based detection, graph-based feature methods, and deep learning techniques. The selection of the detection algorithm depends on the specific requirements of the application and the level of security needed.

Electronic Control Units (ECUs) are an essential component of the Controller Area Network (CAN) protocol used in the automotive industry. ECUs communicate with each other over the CAN bus protocol, which ensures high communication rates. However, the CAN protocol is prone to various cybersecurity attacks, and ECUs are vulnerable to these attacks. To address this issue, researchers have proposed ECU fingerprinting algorithms to detect and prevent cyber-attacks on the CAN bus. Here is an overview of some of the state-of-the-art ECU fingerprinting algorithms:(1)Clock-based IDS (CIDS): CIDS is an anomaly-based intrusion detection system that measures and exploits the intervals of periodic in-vehicle messages for fingerprinting ECUs. The fingerprints are then used for constructing a baseline of the ECUs’ clock behaviors with the Recursive Least Squares (RLS) algorithm. Based on this baseline, CIDS uses Cumulative Sum (CUSUM) to detect any abnormal shifts in the identification of errors, which is a clear sign of intrusion [50,51,52,53,54,55,56].(2)Physical-Fingerprinting of Electronic Control Unit (ECU) Based on Machine Learning Algorithm: This algorithm uses machine learning algorithms to identify the physical fingerprints of ECUs based on the time and frequency domain features of the consecutive CAN packets. The algorithm classifies the ECUs based on their physical fingerprints and detects any abnormal behavior [57,58,59,60,61,62,63,64].(3)ECU Fingerprinting through Parametric Signal Modeling and Artificial Neural Networks: This algorithm uses parametric signal modeling and Artificial Neural Networks to identify the physical fingerprints of ECUs. The algorithm extracts the features of the CAN packets and uses them to train the Artificial Neural Network. The trained network is then used to classify the ECUs and to detect any abnormal behavior [65,66,67,68,69,70].(4)Two-Point Voltage Fingerprinting: This algorithm uses voltage measurements to identify the physical fingerprints of ECUs. The algorithm measures the voltage at two points in the CAN bus and uses the difference between the two measurements to identify the ECU. The algorithm can detect any masquerading attacks on the CAN bus [71,72,73,74,75,76].
In summary, ECU fingerprinting algorithms are proposed to detect and to prevent cyber-attacks on the CAN bus. These algorithms use different approaches such as clock-based IDS, machine learning algorithms, parametric signal modeling, and Artificial Neural Networks. The selection of the ECU fingerprinting algorithm depends on the specific requirements of the application and the level of security needed.

## 2. Background on CAN Cybersecurity

### 2.1. CAN Protocol Basics

The Controller Area Network, commonly referred to as CAN bus, is a serial standard for field buses that is primarily employed in the automotive industry. It was introduced in the 1980s by Robert Bosch as a means to connect various electronic control units (ECUs). Notably, the CAN protocol offers a range of key advantages:Simplicity of Wiring: The CAN bus operates on a message-oriented approach, rather than an address-oriented one. This design allows for the straightforward addition or removal of peripherals (nodes), simplifying the wiring process.Rigid Response Times: CAN bus technology enables the creation of systems with highly predictable and rigid response times. This is achieved through specific techniques that are designed to minimize time-related delays.High Immunity to Interference: The ISO 11898 standard mandates that the CAN protocol must maintain operability, even in scenarios where one of the two wires is severed, or if a bus line to the power supply experiences a short-circuit.Error Confinement: Each peripheral device connected to the CAN bus possesses the capability to self-diagnose hardware issues. In the event of a malfunction, a peripheral can voluntarily remove itself from the bus, allowing other peripherals to continue using it.Multi-Master Protocol: Within the CAN protocol, every node has the capacity to compete for control of the bus. This means that each node can assume the role of a master, taking control of the bus and initiating transmissions.
To manage traffic on the bus effectively, the CAN protocol employs the CSMA/BA (Carrier Sense Multiple Access/Bit-wise Arbitration) method. In situations where two or more nodes attempt to transmit simultaneously, an arbitration mechanism based on priority is applied.

### 2.2. Vulnerabilities and Attack Scenarios

It is brought to attention that the CAN bus, an essential communication protocol in various automotive systems, is deficient in fundamental security measures, rendering the wired units susceptible to potential breaches orchestrated by malevolent entities. According to the CIA (Confidentiality, Integrity, Availability) security model, a comprehensive examination reveals the existence of six critical vulnerabilities within the CAN bus framework. These vulnerabilities emerge from two distinct sources: the vulnerabilities concerning the traffic transmission through the CAN bus and those intrinsic to the protocol’s unique characteristics [77,78,79,80].

Among the pressing concerns, the absence of encryption, authentication, and integrity checking in the data transmission via the CAN bus represents a severe violation of the fundamental principles of data security, particularly confidentiality and integrity. Furthermore, the characteristics inherent in the CAN bus protocol, such as broadcast transmission, priority-based arbitration, and limited bandwidth, contribute to the system’s susceptibility to various security threats. The combination of these factors contributes to the heightened risk of a Denial-of-Service (DoS) attack, thus compromising the system’s availability. The specific vulnerabilities identified within the CAN bus context can be discerned as follows:The lack of encryption allows potential adversaries to decipher the historical data transmitted via the CAN bus, thereby comprehending the intricate functionalities of the target Electronic Control Units (ECUs) with relative ease [81,82].The absence of an authentication mechanism in the CAN frame implies that any transmitter can surreptitiously send deceptive CAN frames to any of the interconnected ECUs, potentially gaining unauthorized control over the target ECUs [83,84,85].The absence of integrity checking exacerbates the security concerns, as the receivers might unknowingly accept manipulated data, leading to potential system malfunctions or even complete breaches by malevolent entities [86,87].The broadcast transmission characteristic of the CAN bus, where the frames are disseminated to all interconnected ECUs, acts as a double-edged sword, facilitating system-wide communication, but also enabling unauthorized eavesdropping, which jeopardizes the confidentiality of the communication.The priority-based arbitration, which allows frames with higher priority to dominate the communication channel, poses a significant security risk, as it enables an aggressive Electronic Control Unit (ECU) to manipulate the communication channel, potentially disrupting the entire network’s functioning [88,89,90].The limited bandwidth and payload capacity of the CAN bus results in the insufficiency of robust access control mechanisms, creating a vulnerability that could be exploited by adversaries attempting to compromise the security of the system.

The collective presence of these vulnerabilities within the CAN bus infrastructure calls for urgent attention to fortify the security measures and to establish robust protocols to safeguard against potential breaches and malicious attacks that could compromise the integrity and functionality of the system. In the following, we also report on the definition of specific cyber-attacks that could be applied on the CAN base networking system.
(1)Unauthorized access: Since the network is centralized, nodes trust each other, and a malicious node that is attached to the network can have access to all the data flowing and can disrupt the data flow [91,92,93,94,95,96,97].(2)Replay attacks: An attacker intercepts and records a message, and then replays it later to achieve a malicious goal [98,99,100,101,102].(3)Denial of Service (DoS) attacks: An attacker can flood the network with messages, causing the network to become unresponsive [103,104,105,106,107].(4)Spoofing attacks: An attacker can send messages with a fake source address, making it difficult to identify the source of the attack [108,109,110,111,112].(5)Physical layer attacks: An attacker can manipulate the physical layer of the CAN bus to cause malfunctions in CAN nodes [113,114,115,116,117,118].
To address these vulnerabilities, various solutions have been proposed, such as intrusion detection systems, encryption, and authentication mechanisms. However, there is no optimal solution, and the problem is mitigated with network segmentation and intrusion detection systems. It is essential to establish a strong security system for automotive networks to maintain the advances in safe technologies and to advance the state of the art in automotive cybersecurity [119,120,121,122].

To monitor message flow from different ECUs, a modern CAN-based network can be accessed by peripherals like Bluetooth, Wi-Fi, and OBD. This makes it possible for IDs to be replicated, which can prevent some ECUs from communicating. Different vulnerabilities exist based on the hardware, software, and attack surfaces of the ECUs in the CAN network; the idea of Strong and Weak Attackers is explained. Fully and weakly compromised ECUs are the two categories of compromised ECUs that we distinguish. A weakly exploited ECU lacks the capacity to insert fake messages, and can stop some message transmissions or function in listen-only mode. On the other hand, an attacker with complete access to an ECU can take full control, access data stored in memory, and insert any attack message. Because the CAN bus protocol does not provide encryption, authentication, or integrity checking, it is vulnerable to a number of security issues. The system is unable to determine whether the data have been replayed by a malicious node, even in the event that cryptographic techniques are used.

We consider three main attack paths based on these weaknesses. Because integrity checking is not present, the impersonation attack can change CAN frames, and the replay attack can succeed if sufficient defenses are not taken.

Replay Attack for CAN: Without authentication and integrity for the CAN frames, a Strong Attack is able to launch the replay attack. As shown in Figure 1, a fully compromised ECU A transmits the CAN frames received from the ECU C, modifying its data field. As a result, the receiver ECU B will function abnormally under the replayed control information.Impersonation Attack for CAN: Having known the IDs of the CAN frames from ECU B, the Strong Attack is able to launch the impersonation attack, as shown in Figure 2. The Weak Attacker first suspends the transmission of ECU B, and the strong attacker then controls ECU A to transmit the CAN frames using the ID of ECU B to manipulate the target, ECU C.Injection Attack for CAN: As shown in Figure 3, a Strong Attacker ECU A is able to inject CAN frames with arbitrary IDs and content. On the one hand, the injected frames with the highest priority ID will always occupy the CAN bus. On the other hand, it can compromise the functionality of the bus occupying the transmission.

## 3. Proposed Algorithm Design

The primary objective of this research is to demonstrate the deployment of a classification system designed for ECUs that are connected to the CAN network. This system leverages the NXP S32K144 embedded system as a Traffic Analyzer. The classification process relies on fingerprinting features and is executed through a pre-trained neural network.

### 3.1. Voltage Sampling Method

The objective here is to identify a sampling technique that is capable of optimizing the performance of the ADC integrated into the S32K144 board, which serves as the Traffic Analyzer. The goal is to achieve the highest possible number of voltage samples at a 12-bit resolution. This is accomplished by utilizing the Hardware Trigger mechanism in conjunction with the PDB timer module, as illustrated in Figure 4.

This approach significantly boosts the sampling rate, achieving a five-fold increase compared to the Software Trigger method, which is typically adopted in embedded systems. With the ADC Hardware Trigger method, the PDB timer module is employed to initiate ADC conversions, enabling the conversion of analog voltage inputs from two distinct channels, namely CANH and CANL, into digital values. Given the specified parameters:-Bit resolution = 12 bits-CAN rate = 125 Kbit/s-Bit number for message = 110 bits-PDB Period = 2.15 μs
We can calculate the following:(1)One message time:
–One message time = (Bit number per message)/(Can bus velocity)–One message time = 110 bits/(125 Kbit/s) = 880 μs(2)Number of samples per each message:
–Number of samples per each message = (One message time)/(PDB period)–Number of samples per each message = 880 μs/2.15 μs ≅ 410 samples
This calculation is performed for each channel, resulting in a total of 820 samples.

### 3.2. Features Extraction

The voltage features represent the measurable characteristics of the phenomenon under observation. Only dominant values are taken into consideration for feature calculation because they correspond to the moments when the units transmit voltage values. Values associated with the ACK bit are excluded from consideration as they signify the instances where each of the ECUs acknowledges the receipt of the message. To illustrate this, consider the sampling of a CAN signal from a message transmitted on the bus, as depicted in Figure 5. In this context, dominant values are graphically identified as those lying above the average voltage of CANH and below the average voltage of CANL.

The dominant voltage samples that we acquired are used for extracting features. However, we chose to utilize only six out of the twelve features that were initially proposed. These features are divided equally between CANH and CANL, resulting in a total of twelve features. While we did explore the use of frequency-based features, they were found to be impractical given the limited number of dominant samples that can be obtained from each message. In Table 1, the features used as input for the proposed Artificial Neural Network classifier are reported.

### 3.3. Features Scaling

In this section, our goal is to establish continuous communication among the three units (ECUs). Meanwhile, the Traffic Analyzer will print feature values that are associated with the sender for each message. We anticipate sending approximately 1000 messages on the bus using the communication method illustrated in Figure 6.

After collecting the data, we proceed to analyze the data trends for the three units. We compare the Probability Density Functions (PDFs) estimated from the features obtained from both CANH and CANL to a Normal distribution. The Normal distribution is characterized by a mean that is equal to the mean of the analyzed feature, and a standard deviation that is equal to the standard deviation of the analyzed feature. An example of the data trend for Unit A CANH is illustrated in Figure 7.

The data trends for the other three units exhibit similar patterns to those presented. It is important to note that these data trends do not follow a Normal distribution. In machine learning, it is a common practice to scale input data for neural networks to eliminate redundancy, enhance stability, and facilitate convergence. Given the non-Gaussian distribution of the data, we opted for Normalization using the Min-Max scaling method rather than Standardization for feature scaling.
(1)$$X^{\prime }=\frac{X-X_{\min}}{X_{\max}-X_{\min}}$$

### 3.4. Neural Network

It has been determined that the most suitable approach to implementing a neural network involves use of the TensorFlow [123] and Keras [124] environments. This choice offers the advantage of allowing for the use of the TensorFlow Lite format, which in turn allows us to exploit the capabilities of the hardware while reducing the size of the network in terms of storage space, measured in Kbytes. The characteristics of the chosen neural network model are described below:Learning Algorithm: The learning algorithm selected for classification is a Supervised Learning Algorithm. In particular, Gradient Descent is a common technique that is used to optimize the weights of the neural network during the training process. This algorithm can be implemented using several variations, including Stochastic Gradient Descent (SGD), which uses a random sample to calculate the weight update, and Adaptive Gradient Algorithm (adagrad), which adapts the learning rate for each parameter of the network. The calculation of stochastic gradient descent occurs according to Equation (Equation 2):
(2)$$w_{t+1}=w_{t}-\alpha \nabla Q\left(w_{t}\right)$$
where $w_{t}$ represents the weights of the network at time *t*, α is the learning rate, and $\nabla Q(w_{t})$ indicates the gradient of the cost function *Q* with respect to the weights $w_{t}$.Activation Function: The Rectified Linear Unit (ReLU) activation function was chosen, defined as f(x)=max(0,x). ReLU is one of the most widely used activation functions for hidden layers of neural networks. Its simplicity of implementation and compatibility with TensorFlow Lite makes it a practical choice.Model Type: The type of neural network model adopted here is a Feed-Forward network. In this type of model, connections exist only between successive levels, avoiding interconnections between neurons of the same level.Optimization Algorithm: The Adam optimization algorithm, derived from adaptive moment estimation, was selected. Adam is an extension of Stochastic Gradient Descent (SGD), which combines first-order and second-order information to update weights efficiently, and with an adaptive learning rate. The weight update rule in Adam is defined with the set of recursive equations in Equation (Equation 3):
(3)$$\begin{array}{c} m_{t+1}=\beta_{1}m_{t}+\left(1-\beta_{1}\right)\nabla Q\left(w_{t}\right)\\ v_{t+1}=\beta_{2}v_{t}+\left(1-\beta_{2}\right)\nabla Q{\left(w_{t}\right)}^{2}\\ w_{t+1}=w_{t}-\alpha \frac{m_{t+1}}{\sqrt{v_{t+1}}+\epsilon } \end{array}$$
where $m_{t}$ and $v_{t}$ represent the moment and second moments of the gradient at time *t*, respectively; and $\beta_{1}$, $\beta_{2}$, and ϵ are hyper-parameters of the model.Output Function: For the output layer of the neural network, the Softmax function was chosen and it is defined as reported in Equation (Equation 4).
(4)$$\sigma \left(z_{j}\right)=\frac{e^{z_{j}}}{\sum_{k=1}^{K}e^{z_{k}}}$$
where $z_{j}$ represents the function input and *K* is the total number of classes. Softmax is a mathematical function that transforms a vector of numerical values into a vector of probabilities. Each probability in the output vector corresponds to the relative scale of the corresponding input value.

This combination of elements within the neural network model aims to facilitate effective classification and prediction tasks, making use of widely accepted practices in the field of deep learning. Figure 8 shows the internal architecture of the proposed neural network classifier.

We want to emphasize that the Feed-Forward neural network was chosen to reduce the problems of the interpretability of results, which in fact remains a key issue in industrial applications, and is in fact the weak point of AI models that rely on much more sophisticated learning paradigms.

In fact, data from the physical layer are statistically processed using filtering and feature extraction/selection techniques. Since the features have been analyzed and selected a priori, the network is not left with the task of extracting features from the data, as has been achieved with the Deep Learning approach, for example, but only to recognize the interconnections between the class and the features. This reduces the problems of the interpretability of the results compared to using an AI model without the data manipulation steps, as it is not left to the network to decide which features represent the physical data associated with each control unit. In addition, this greatly reduces the size of the neural network model, which can then also be integrated into embedded systems with small computational and memory resources.

### 3.5. TensorFlow Lite for Embedded Integration

The primary objective of this section is to successfully implement neural network algorithms on the S32K144 microcontroller, which possesses limited computational and memory capabilities. To address these constraints effectively, we have employed TensorFlow Lite, a specialized toolkit designed for optimizing and deploying machine learning models on embedded and IoT devices. TensorFlow Lite offers two core tools: the TFLite Converter and the TFLite Interpreter. The primary role of the Converter is to enhance the model’s performance by reducing its size and improving its execution speed. This optimization primarily hinges on a fundamental technique known as model quantization, wherein all weight values are converted from the standard 32-bit floating-point format to 8-bit integers (post-training quantization). While this quantization process may introduce some slight trade-offs in terms of model accuracy, it significantly reduces the model’s size, making it more lightweight and responsive.

Once the model has undergone conversion into the TensorFlow Lite format, the Interpreter, which is deployed on the embedded system, can be invoked to perform inference tasks. Notably, during the time when these tests were conducted, there was no official TensorFlow Lite support available for the NXP S32K144 microcontroller. Consequently, it became imperative to devise a method, as outlined in Figure 9, for importing TensorFlow Lite libraries onto the S32K144 board. This essential step was pivotal in ensuring the seamless integration and functionality of the machine learning model on the microcontroller despite the absence of native support.

## 4. Experimental Validation

In this section, we delve into the implementation of an anomaly detection algorithm tailored for the CAN bus, leveraging Artificial Neural Networks (ANNs). To establish a realistic testing environment, we have engineered a circuit that emulates the structure of an actual CAN network or sub-network. This circuit comprises five Electronic Control Units (ECUs) alongside the Traffic Analyzer, strategically designed to simulate various aspects of CAN communication. Among these ECUs, we have employed five distinct types, each equipped with one of four different types of CAN communication modules. Within this ensemble, Unit A, Unit B, and Unit C serve as our known units. These units play a pivotal role in the training process of the neural network, enabling it to learn and to establish baseline patterns.

On the other hand, we have designated Intruder 1 and Intruder 2 as our unknown units. These units emulate potential intruders within the CAN network, mimicking the behaviors of unauthorized or anomalous entities.

The primary objective here is to develop an anomaly detection system that can effectively identify and flag these intruder units based on deviations from established normal behavior patterns. This comprehensive setup (see Figure 10) allows us to assess the algorithm’s ability to discern between known and unknown units, ultimately enhancing the security and integrity of the CAN network. Given that Unit C and Intruder 2 are identical units in terms of their internal configurations, it naturally follows that their waveforms exhibit a striking degree of similarity. This similarity arises from the shared characteristics and behaviors inherent to these two units. Consequently, their waveforms, when observed, closely mirror each other due to their analogous CAN bus output patterns and operational tendencies.

In stark contrast, the waveform generated by Intruder 1 presents a notably distinct profile in comparison to its counterparts. The distinctive nature of Intruder 1’s waveform is primarily attributed to the unique characteristics of its CAN bus output. Notably, Intruder 1’s CAN bus output exhibits voltage levels that fluctuate within the range of 1 V to 3 V. Importantly, this voltage range aligns perfectly with the established CAN bus protocol standards of 1.5 V to 3.5 V. Therefore, Intruder 1’s waveform conforms to the specified voltage parameters defined by the protocol, albeit with a distinct operational pattern that sets it apart from the other units (see Figure 11).

After successfully loading the trained neural network onto the S32K144 board (specifically, the Traffic Analyzer), a comprehensive test was conducted. This test encompassed the analysis of a total of 1000 messages, with each unit being subjected to a set of 200 messages. The primary objective of this test was to assess the effectiveness of the classification system. In establishing a criterion for classifying units, a conservative approach was adopted. It was determined that a threshold of 90% would serve as the minimum precautionary threshold for class membership. Units that exhibited classification scores equal to or exceeding this threshold would be confidently regarded as belonging to a specific class. However, in cases where units yielded classification scores falling below the 90% threshold, they would be considered as not definitively belonging to any particular class. This approach allowed for a robust classification mechanism that prioritized high confidence in unit assignment, ensuring that any classification made met a stringent threshold of reliability.

The classification of the known units has yielded correct results, effectively categorizing the messages as expected. However, it is worth noting that the neural network, when faced with messages from the unknown units, consistently classifies them as originating from Unit C. This outcome suggests that the neural network, which was trained on data from the known units, is likely recognizing similarities between the unknown units and Unit C’s waveform patterns. A proposed solution to address the challenge of classifying the unknown units involves the creation of a distinct fourth class that encompasses all instances associated with the unknown units.

To facilitate this, a fictitious dataset was meticulously generated, comprising a total of 3000 observations, each consisting of 200 dominant values for each channel. This dataset was custom-built using Matlab, employing the following equations:(5)$$\begin{array}{c} \mathrm{Rand}\_\mathrm{dominant}\_\mathrm{CANH}=Randi\left[2900,4500\right]+N\left(\mu_{h},\sigma_{h}\right)\\ \mathrm{Rand}\_\mathrm{dominant}\_\mathrm{CANL}=Randi\left[500,2000\right]+N\left(\mu_{l},\sigma_{l}\right) \end{array}$$

Figure 12 represents the training and validation phases of the selected neural network classifier in the configuration for the first test, while Table 2 reports on the results obtained regarding accuracy in classification.

Figure 13 represents the dummy physical layer created specifically to retrain the neural network to associate the physical layer of external devices (such as intruder 1 and intruder 2) with an Unk class and to prevent it from being confused with one of the ECUs on which the training was performed.

Here, the function Randi is utilized to generate uniformly distributed pseudo-random integers within a specified range. Using these fictitious values, additional features will be computed and subsequently integrated into the actual dataset. This process will culminate in the creation of a final dataset, consisting of a total of 12,000 observations. This includes the original 9000 real observations, complemented by an additional 3000 fictitious ones. In contrast to the previous training sessions, the learning curve in this case exhibits a distinct trend, although it ultimately converges to an accuracy value of 1. Employing the same evaluation criteria as in the initial test, the results of a test involving 1000 messages (with 200 messages from each unit) are presented below. Each message sent by Intruder 1 was consistently classified as originating from an Unknown unit, indicating that the algorithm was able to effectively distinguish this intruding unit. In contrast, the majority of messages from Intruder 2 remained unclassified, suggesting that the neural network had difficulty assigning them to a specific class or category. Figure 14 illustrates the learning and validation behaviors of the neural network classifier.

In Table 3, the results obtained during the second test are shown, where the neural network is able to associate anomaly detection with the "Unknown" class. It must be highlighted that for Intruder 2, it that seems that a low rate of accuracy in classification occurs, along with a high rate of non-classified observations (188 over a total of 200). This is due to the threshold of 90% selected in the output to classifier. This means that the neural network provides as an output a vector with similar estimated probability (by the softmax layer) for the possible class. With a lower threshold, it is possible to increase the accuracy, but also with a higher rate of false-positive estimations.

### 4.1. Anomaly Detection Strategy

Building upon the preceding findings, it is conceivable to devise an algorithm that is capable of distinguishing between an attempted attack by an external entity, and a compromise of one or more ECUs within the network. This algorithm, which takes as its input the output values generated by the Softmax function, operates upon the following premise: if the highest score among the first three classes is 90% or greater, it categorizes the message as originating from an Internal Unit; otherwise, it designates it as stemming from an External Unit. Furthermore, the algorithm leverages its knowledge of the ID map that each Unit is capable of transmitting. With this information, the algorithm gains the capability to determine whether a unit is employing messages with its designated ID or is employing other IDs. This additional layer of analysis enhances the algorithm’s capacity to differentiate between legitimate internal communications and potential external intrusions. See Table 4 for the ID configuration within the proposed validation tests.

The algorithm exhibits the ability to discern four distinct categories of anomalies, thereby enhancing the overall security of the system:External Signal with an Internal ID: In this situation, the algorithm classifies the incoming message as originating from an unknown unit. Remarkably, the message ID aligns with one of those previously loaded onto the ID map. This occurrence suggests a potential external intrusion into the system, as the message source is not recognized as any of the legitimate internal units.External Signal with an External ID: When the algorithm categorizes the message as belonging to an unknown unit, it further scrutinizes the message ID. In the event that the message ID does not correspond to any of the IDs pre-loaded on the ID map, this anomaly is recognized. Such a situation implies the presence of an unauthorized, external source that is trying to communicate with the system.Internal Signal with a Stolen ID: If the algorithm identifies a message as belonging to Unit A, Unit B, or Unit C, and the message ID aligns with one of the IDs available on the ID map, an additional layer of scrutiny is applied. In the case where the source of the message does not match the expected unit, the algorithm flags this as an anomaly. It suggests that an internal but unauthorized unit may be attempting to impersonate a legitimate one.Internal Signal with an External ID: Whenever the algorithm classifies a message as being associated with Unit A, Unit B, or Unit C, it extends its examination to the message ID. If the message ID fails to correspond to any of the IDs pre-loaded on the ID map, the algorithm recognizes this as an anomaly. In such a scenario, it indicates that a message from an internal unit is being sent with an ID that is not recognized by the system, implying an irregularity in the communication protocol.

### 4.2. Attack Simulation

Utilizing the experimental framework at our disposal, we are equipped to replicate the three distinctive forms of attacks outlined previously. Leveraging the anomaly detection algorithm, the system adeptly identifies these attacks and promptly disseminates warning messages on the console to apprise the system administrator of the detected threats.

**Replay Attack**: Within this particular scenario, Intruder 1 assumes the role of a Strong Attacker. Following a period of regular network operations, whenever a message carrying an ID assigned to Unit A is transmitted, the Strong Attacker initiates a response on the bus. These responses are intentionally infused with malicious content bearing the same ID as the legitimate messages transmitted by Unit A. The resulting test outcomes are comprehensively illustrated in Figure 15.

**Impersonation Attack**: In this specific context, Intruder 1 once again adopts the persona of a Strong Attacker. After a phase of routine network activities, Unit A becomes the target of an attack orchestrated by a Weak Attacker. The scheme involves the Weak Attacker temporarily disrupting the transmission of messages from Unit A by placing it in a silent or listen-only mode. Capitalizing on this vulnerability, the Strong Attacker seizes the opportunity to impersonate the compromised unit, transmitting harmful content under its guise. The consequential test outcomes are meticulously represented in Figure 16.

**Injection Attack**: In the context of this simulated situation, Intruder 1 operates as a Strong Attacker, exclusively employing high-frequency messages with the specific ID of [0x00]. The primary objective here is to flood the bus, thereby preempting all arbitration phases and effectively monopolizing the communication medium. Consequently, this impedes any legitimate interaction among the other units. A comprehensive depiction of the results derived from this testing is presented in Figure 17.

These simulated attacks serve as valuable test cases for evaluating the robustness and effectiveness of the Anomaly Detection algorithm under various security threats, enabling the system to proactively respond to potential vulnerabilities. The results obtained with the simulated attacks are perfectly in line with the accuracy obtained during the validation phase of the classifier model shown in the previous section.

### 4.3. Thermal Test for Prediction Robustness

Electronic control units are commonly put in situations that are characterized by volatile temperature variations, often varying by large margins. These systems rely significantly on the CAN bus for communication. When these control units are used in automotive applications, their temperatures might vary depending on a number of circumstances, such as how close they are to the engine, how long they run for, and how exposed they are to outside elements like direct sunlight. Because MOSFETs and resistances are intrinsically sensitive to temperature changes, even small changes in temperature can have a significant impact on the complex integrated circuits present in these devices.

There is a significant degree of danger associated with subjecting these control units to high temperatures, since the heat generated by these conditions can significantly distort the voltage signals, jeopardizing the accuracy of previously recorded data. As such, careful thermal testing is necessary in order to fully evaluate the possible effects of these temperature-induced changes, particularly with regard to the effectiveness of cybersecurity systems that depend on voltage fingerprinting methods. A deeper comprehension of the true implications of the suggested classification approach may be obtained by attentively analyzing the Softmax outputs in connection to temperature changes. This will allow for the development of more intelligent and reliable security mechanisms. To reduce the measured noise effect, values were collected every 200 temperature values and their mean was calculated. The algorithm is shown in Figure 18.

Using the identical configuration as the Second test, the remaining units connected to the bus were set to silent/listen mode. Only Unit C was active in sending messages during the testing process, while the control unit diligently recorded the corresponding data (see Figure 19 for the experimental setup configuration illustration). Commencing at approximately 24 ${}^{{}^{\circ}}$C (ambient temperature) and progressing up to around 83 ${}^{{}^{\circ}}$C (within the Arduino operating temperature range of −40 ${}^{{}^{\circ}}$C to 85 ${}^{{}^{\circ}}$C), a series of 4 messages were systematically transmitted over the CAN network in 5-degree increments, resulting in a total of 48 transmissions monitored by the control unit. The recorded output values from the neural network’s Softmax were collated and are presented in detail in Table 5, accompanied by a visual representation in Figure 20.

The data presented in Table 5 confirm that despite the variations in temperature, there have been no significant alterations that could potentially compromise the validity of the previous results. Delving into the insights offered by Figure 20, it becomes apparent that the predictive trend remains relatively stable within the temperature range spanning from 24 ${}^{{}^{\circ}}$C to 70 ${}^{{}^{\circ}}$C, with the exception of an outlier value peaking at 97.82%. There appears to be a slight downward trend in the scores beyond 70 ${}^{{}^{\circ}}$C; however, it is worth noting that the values consistently remain above the critical 98.4% threshold. Notably, no considerable dips in accuracy were observed throughout the application of the prescribed methodology.

## 5. Conclusions and Future Work

The significant contributions of this research primarily revolve around the development and enhancement of an embedded anomaly detection system utilizing the NXP S32K144 platform. The methodology employed in this study is rooted in a completely experimental approach, commencing with rudimentary CAN network setups and gradually evolving to more intricate scenarios featuring five Electronic Control Units (ECUs) and a Traffic Analyzer. A pivotal improvement over the existing methods is the adoption of a sophisticated voltage sampling technique that is far superior to the Software Trigger mechanism. The selection of neural network characteristics has been meticulously determined through empirical methodologies. A comprehensive statistical analysis of the features extracted from the data has provided profound insights, guiding the preference for Min-Max normalization over Standardization.

Incorporating TensorFlow Lite onto the NXP S32K144 board has enabled the harnessing of cutting-edge tools in the realm of artificial intelligence, effectively unlocking its real-time classification capabilities. To validate the methodology in more complex environments and in the face of potential attacks, a typical CAN network or sub-network scenario was faithfully recreated. To address the challenge of classifying unknown units using the neural network, a novel solution involving the introduction of a Fictitious dataset was proposed. Furthermore, a series of well-documented simulated attacks, inspired by prominent attack methodologies described in the literature, was executed. The efficacy of these attacks was systematically thwarted by the anomaly detection algorithm, thereby affirming its robust functionality. To extend the practical applications of this technology within the automotive sector, the proposed method was rigorously assessed under the conditions of temperature variation. The results underscore the resilience of the methodology to temperature fluctuations, at least within the range of 25 ${}^{{}^{\circ}}$C to 83 ${}^{{}^{\circ}}$C.

Looking forward, the promising results obtained from this research open doors to a host of potential future developments and extensions. Porting from Laboratory Tests to Real Vehicle Implementation: The next phase following laboratory implementation involves the deployment of this technology in a genuine automotive environment to facilitate functional testing. This real-world application could significantly contribute to the field of automotive cybersecurity. Integration with Other Fingerprinting Techniques: The proposed methodology is amenable to integration with other fingerprinting techniques, including time-based fingerprinting methods and various other fingerprinting approaches. Combining multiple fingerprinting techniques could enhance the overall security of automotive networks. Application to Other Protocols: While this research has been focused on CAN environments, the method’s efficacy encourages its application to diverse network types within the automotive and industrial sectors. This broadening of scope could address security concerns in various communication protocols. Comparison with Deterministic Algorithms: In the realm of classification, an avenue for future exploration lies in the comparison of neural networks with deterministic algorithms, such as Decision Trees, to gauge their relative performances. This could provide valuable insights into the strengths and weaknesses of different approaches. Further Resilience Testing: Extending investigations into the resilience of the system to a broader range of temperature variations can provide valuable information about its practicality under various environmental conditions. Additionally, exploring other potential environmental factors such as humidity and electromagnetic interference can further enhance the system’s robustness. Real-Time Anomaly Detection: Efforts can be directed towards achieving real-time anomaly detection capabilities, potentially reducing response times and increasing the system’s effectiveness in mitigating threats.

In conclusion, this research not only validates the effectiveness of the proposed methodology, but also outlines a promising path for future research, development, and practical implementation in the domains of automotive and industrial cybersecurity. The potential for enhancing network security, particularly in the context of the growing significance of connected vehicles and industrial IoT, makes this work a valuable contribution to the field.

## Figures and Tables

**Figure 1 sensors-23-09231-f001:**
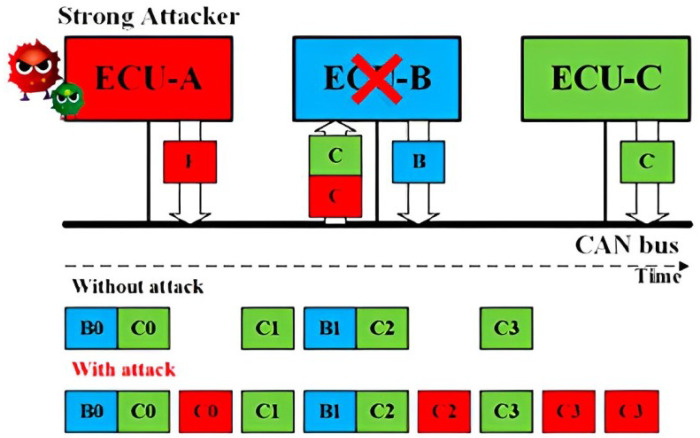
Schematic representation of the Replay attack concept.

**Figure 2 sensors-23-09231-f002:**
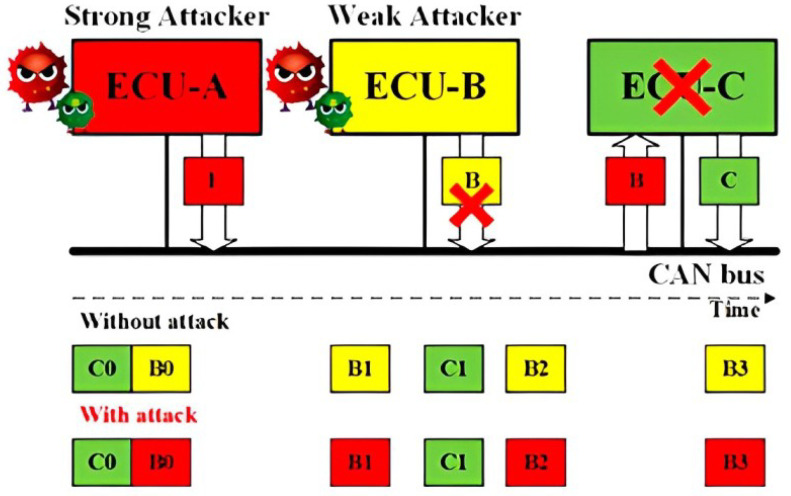
Schematic representation of the Impersonation attack concept.

**Figure 3 sensors-23-09231-f003:**
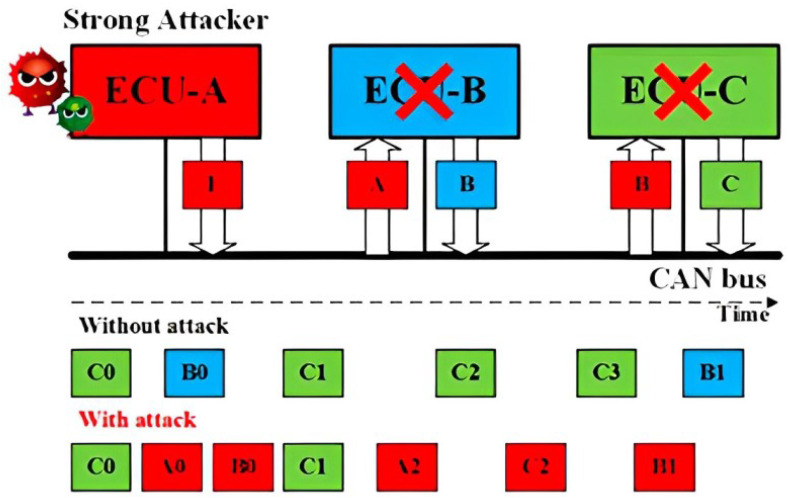
Schematic representation of the Injection attack concept.

**Figure 4 sensors-23-09231-f004:**
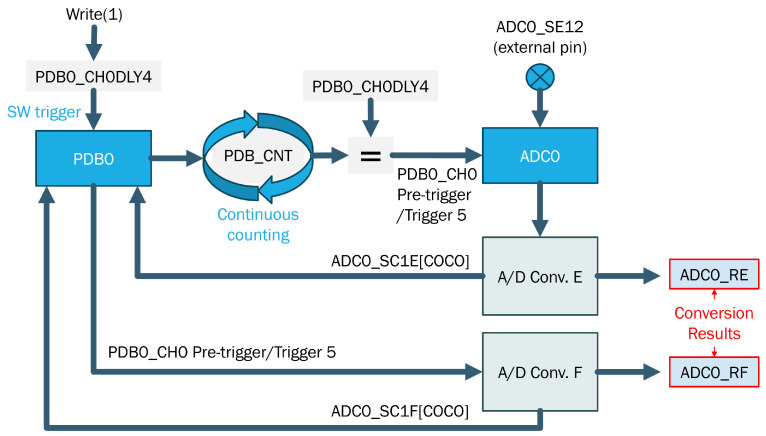
ADC Hardware Trigger with PDB in back-to-back mode.

**Figure 5 sensors-23-09231-f005:**
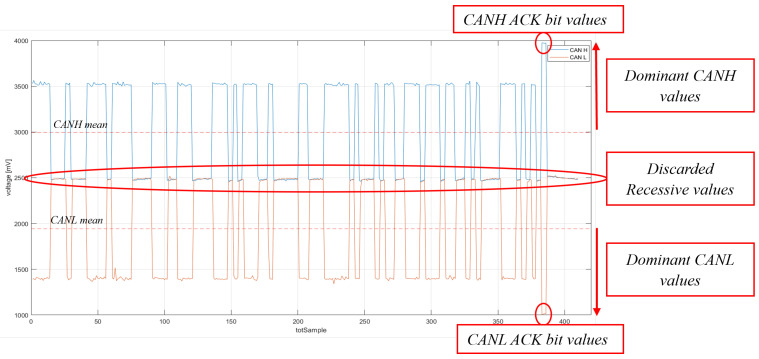
Dominant values extraction.

**Figure 6 sensors-23-09231-f006:**
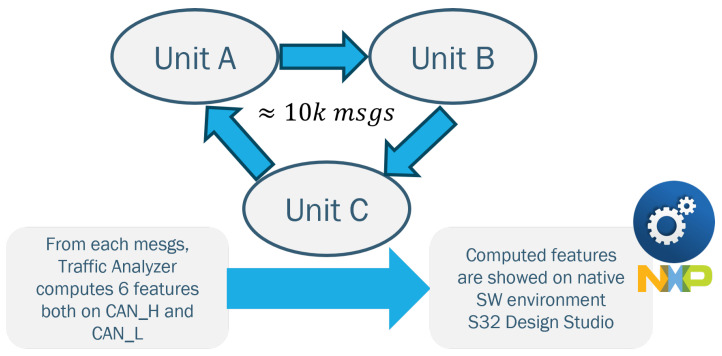
Method of communication between boards during dataset collection.

**Figure 7 sensors-23-09231-f007:**
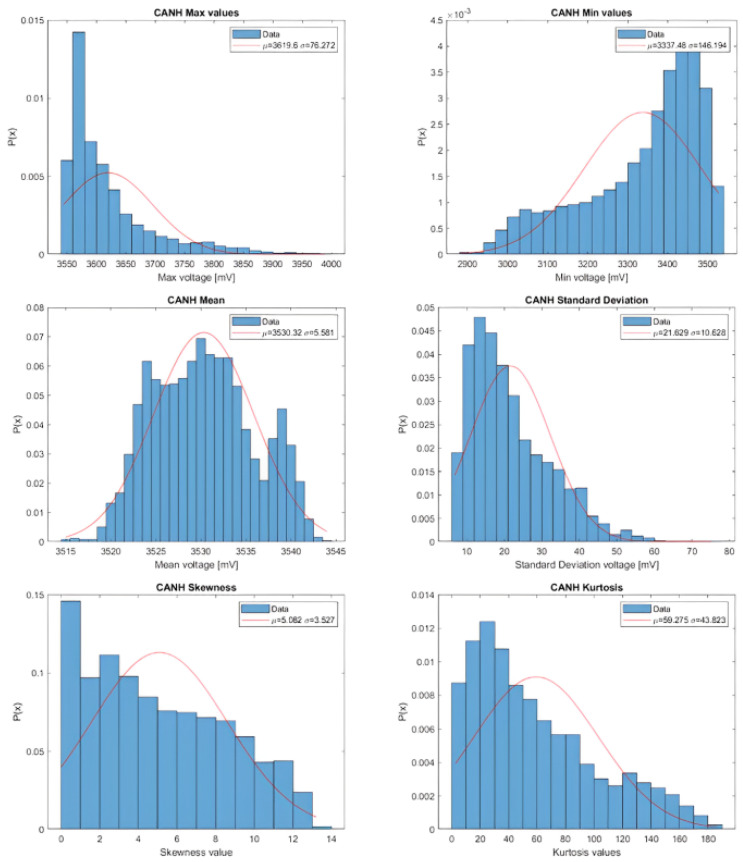
Estimation of Probability Density function of Unit A CANH features compared with the Normal Distribution.

**Figure 8 sensors-23-09231-f008:**
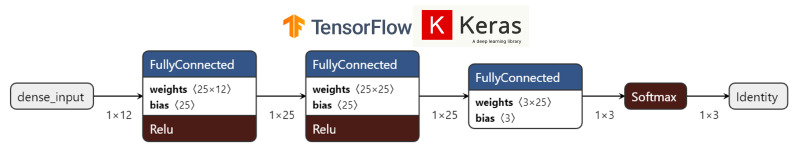
Neural Network with two hidden neural layers of 25 neurons each.

**Figure 9 sensors-23-09231-f009:**
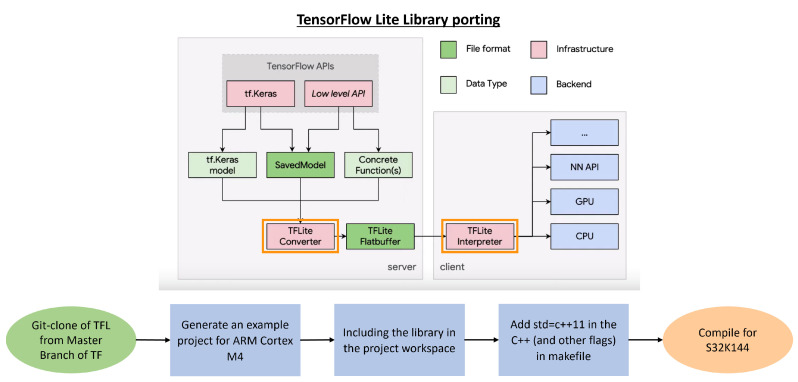
Flow chart of TensorFlow Lite porting Library.

**Figure 10 sensors-23-09231-f010:**
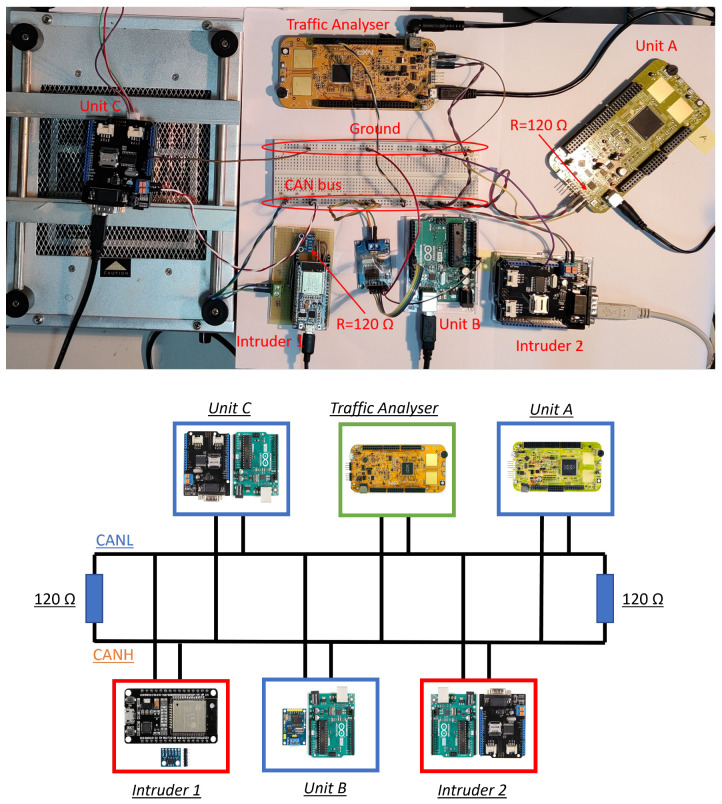
Experimental Setup CAN circuit.

**Figure 11 sensors-23-09231-f011:**
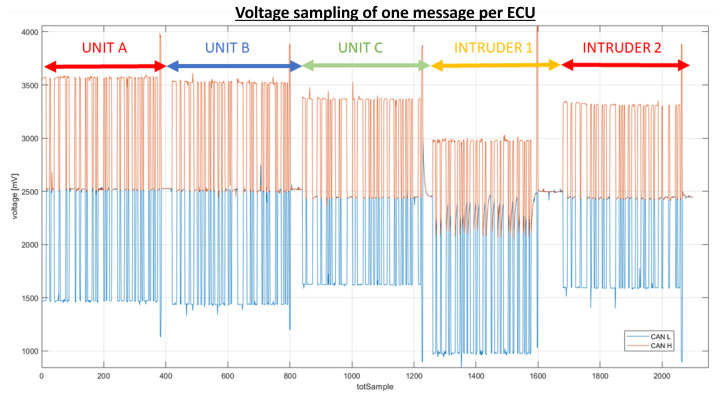
Voltage sampling of one message per unit.

**Figure 12 sensors-23-09231-f012:**
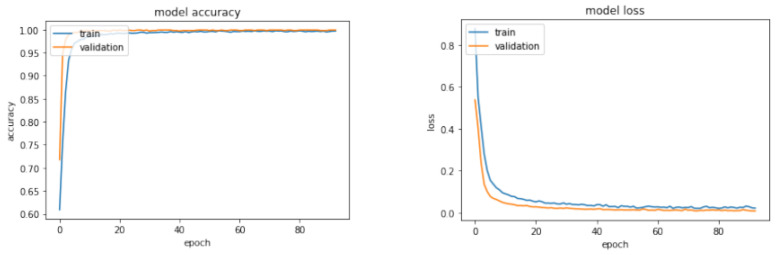
Learning curve of accuracy (**left**) + Learning curve of loss (**right**) in the First Test.

**Figure 13 sensors-23-09231-f013:**
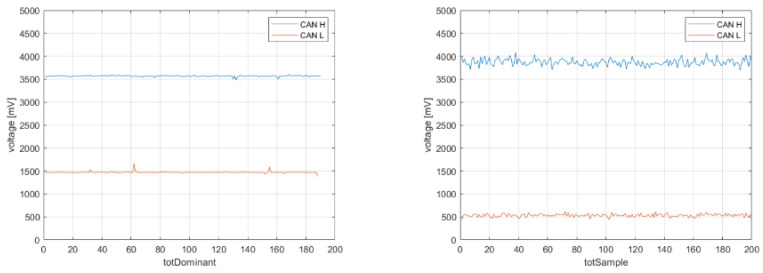
Example of Dominant Real Value (**left**) + Example of Dominant Fictitious Values (**right**).

**Figure 14 sensors-23-09231-f014:**
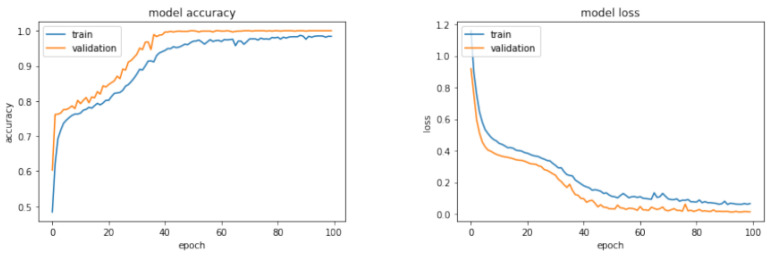
Learning curve of accuracy (**left**) + Learning curve of loss (**right**) for the Second Test.

**Figure 15 sensors-23-09231-f015:**
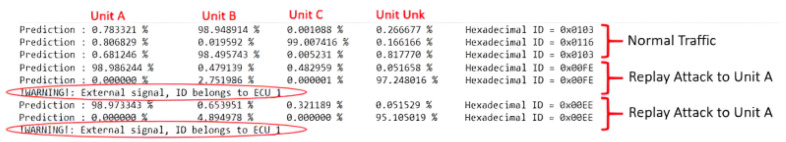
Detection of the Replay Attack.

**Figure 16 sensors-23-09231-f016:**
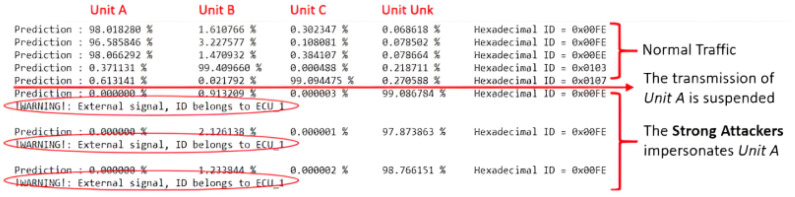
Detection of the Impersonation Attack.

**Figure 17 sensors-23-09231-f017:**
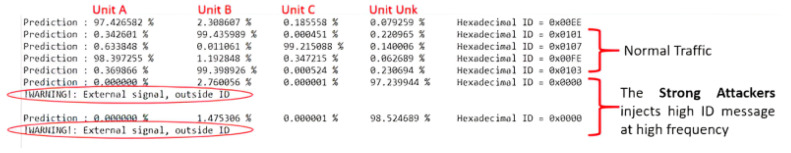
Detection of the Injection Attack.

**Figure 18 sensors-23-09231-f018:**
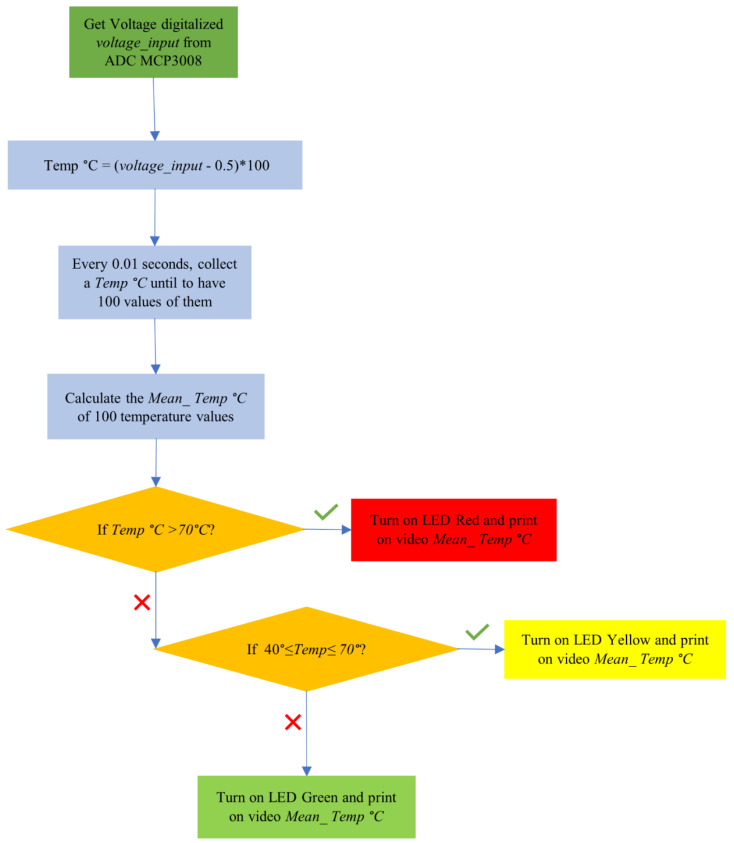
Thermal test procedure deployed on RaspberryPi3B+.

**Figure 19 sensors-23-09231-f019:**
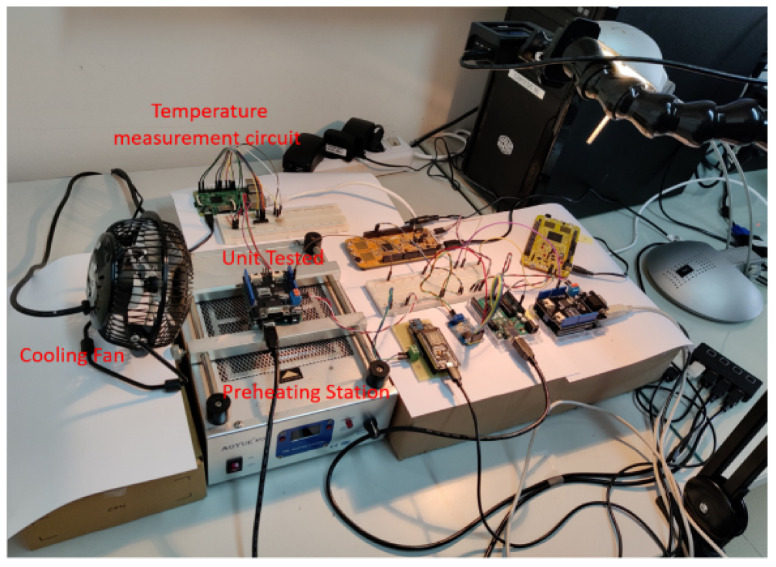
Final Setup for Thermal Test.

**Figure 20 sensors-23-09231-f020:**
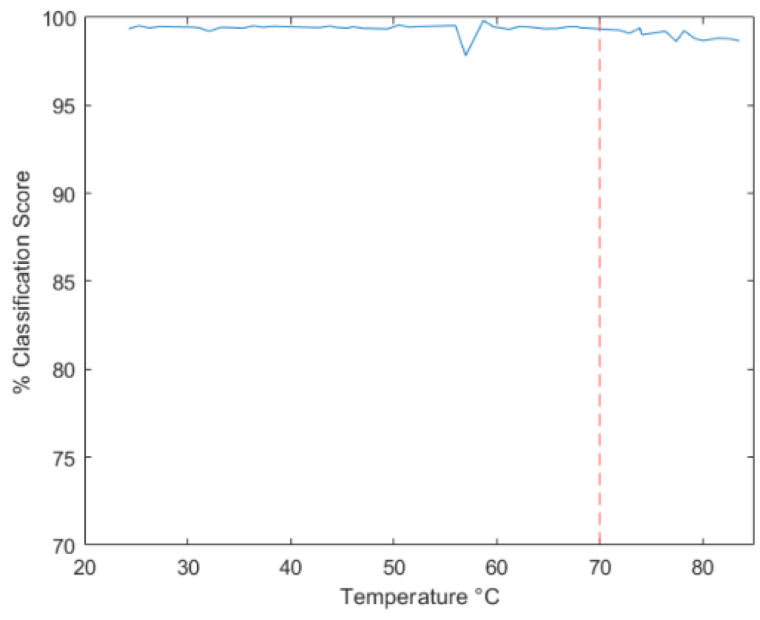
Classification result as a function of temperature.

**Table 1 sensors-23-09231-t001:** Time-domain features set.

Features	Equation
Max value	$M=\max\left(v_{i}\right)$
Min value	$m=\min\left(v_{i}\right)$
Mean	$v_{m}=\frac{1}{n}\sum_{i=1}^{n}v_{i}$
Standard Deviation	$\sigma =\sqrt{\frac{\sum_{i=1}^{n}{\left(v_{i}-v_{m}\right)}^{2}}{n}}$
Skewness	$S=\frac{\sum_{i=1}^{n}{\left(v_{i}-v_{m}\right)}^{3}}{n\sigma^{3}}$
Kurtosis	$K=\frac{\sum_{i=1}^{n}{\left(v_{i}-v_{m}\right)}^{4}}{n\sigma^{4}}$

**Table 2 sensors-23-09231-t002:** Classification results on 200 messages in First Test.

	Number of Times Softmax Score ≥90%		
	Classified as A	Classified as B	Classified as C
Unit A	200	0	0
Unit B	0	200	0
Unit C	0	0	200
Intruder 1	0	0	200
Intruder 2	0	0	200

**Table 3 sensors-23-09231-t003:** Classification results on 200 messages in Second Test.

	Number of Times Softmax Score ≥90%			
	Classified as A	Classified as B	Classified as C	Classified as Unk
Unit A	191	0	0	1
Unit B	0	200	0	0
Unit C	0	0	200	0
Intruder 1	0	0	0	200
Intruder 2	0	0	10	2

**Table 4 sensors-23-09231-t004:** ID configurations for testing.

Unit	ID HEX	Standard Frame	Data Length [Bytes]	Randomized Data [Bit]	Bit Rate
A	EE, FE	Yes	8	64	$125\frac{\mathrm{kbit}}{s}$
B	101, 103	Yes	8	64	$125\frac{\mathrm{kbit}}{s}$
C	105, 116	Yes	8	64	$125\frac{\mathrm{kbit}}{s}$

**Table 5 sensors-23-09231-t005:** Unit C Classification results during Thermal Test on 48 message.

	Unit C			
	Classified as A	Classified as B	Classified as C	Classified as Unk
Mean score	0.395120%	0.013482%	99.328014%	0.263384%
Std Dev score	0.103029%	0.014331%	0.258574%	0.233061%
Number of times score ≥75%	0	0	48	0
Number of times score ≥90%	0	0	48	0

## Data Availability

Data are contained within the article.

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
