# Peer review of "Design and Experimental Assessment of Real-Time Anomaly Detection Techniques for Automotive Cybersecurity"

_sensors, 2023, doi:10.3390/s23229231_

Round 1
Reviewer 1 Report
Comments and Suggestions for Authors
Dear Authors, congratulations for the work and contribution. However, there are some questions to be done:
- You don't need to put Fig. 1 because it is well known
- There are more CAN vulnerabilities. Please, check out.
- Take a look in terms of the CAN Attacks Scenarios
- How does the attacker interfer on the CAN signal?
- I didn't identify the attacker intention
- Why and how did you use the ANN?
- Good experimental setup and just need to be better explained.
- Describe better Fig. 10
- Improve data in Table III and IV
- What was the data acquisition system you used? Try to clarify it.
- What is the proposal of thermal test?
- Please, describe and present the countermeasures.
Reviewer 2 Report
Comments and Suggestions for Authors
Overall, well written paper with aspects of engineering.
In case to prove the scientific input among other researches, 1 section requires update. It is not clear what kind of results were achieved by other authors. Basically, current "Related works" does not reflect itself. Please provide results in values of the described papers where is possible. Then, table with identified parameters might be added which leads to the need of further i.e. yours research.
Reviewer 3 Report
Comments and Suggestions for Authors
The research goals solving the problems of cybersecurity are well defined and justified.
The targeted applications concern automotive industry that use the Controller Area Network (CAN) protocol implemented on some Electronic Control Units (ECUs).
The paper proposes as novelty an ECU recognition algorithm, implemented on a pre-trained neural network, used for (malicious) intruders’ detection.
The approach is well linked to the domain’s literature of the last years mentioning good and relevant references. A good state of the art for CAN anomaly detection is given, The paper contains the CAN bus networking, its vulnerabilities and some possible attack scenarios are discussed.
Some tests and their analysis are are presented and discussed. They are convincingly sustaining the proposed methods and procedure.
The research methodology and the application development are provided in a clear and good intuitive manner using some figures that help the understanding.
Even if the methodology and the application development are well explained and analyzed, I would recommend the paper improvement using some formal mathematical descriptions. I suggest linking the formulas with the methods.
Another recommendation: please explain the results of Figure 2. Bottom.
In the Introduction chapter is a declaration that the proposed methodology leverages real-time analysis on a platform of low complexity. The temporal requirements are not clearly specified, and their formal verification is not provided.
